# Robust Training of Neural Networks at Arbitrary Precision and Sparsity

## Abstract

The discontinuous operations inherent in quantization and sparsification introduce obstacles to backpropagation. This is particularly challenging when training deep neural networks in ultra-low precision and sparse regimes. We propose a novel, robust, and universal solution: a denoising affine transform that stabilizes training under these challenging conditions. By formulating quantization and sparsification as perturbations during training, we derive a perturbation-resilient approach based on ridge regression. Our solution employs a piecewise constant backbone model to ensure a performance lower bound and features an inherent noise reduction mechanism to mitigate perturbation-induced corruption. This formulation allows existing models to be trained at arbitrarily low precision and sparsity levels with off-the-shelf recipes. Furthermore, our method provides a novel perspective on training temporal binary neural networks, contributing to ongoing efforts to narrow the gap between artificial and biological neural networks.

## 1 Introduction

The recent surge in the size and complexity of generative AI models has elevated computational efficiency to the forefront of AI research (Chowdhery et al., 2022; Peng et al., 2023). Among the diverse approaches to achieving efficiency, quantization and sparsification techniques stand out as two classic and widely explored methods. Quantization and sparsity techniques can effectively reduce the computational requirements of large language models (LLMs). Quantization reduces the precision of model weights and activations, thereby decreasing storage requirements. Sparsification, on the other hand, prunes redundant weights, leading to a more compact model. These techniques enable LLMs to be deployed on resource-constrained devices, such as mobile phones and embedded systems, while also improving their speed and memory efficiency. This facilitates the widespread adoption of LLMs by a broader range of individuals and businesses.

Despite their promise, quantization and sparsification introduce non-differentiable operations, such as rounding and hard thresholding, which are incompatible with the differentiable design of backpropagation, the cornerstone of neural network training. This incompatibility has plagued training algorithms for decades, hindering progress in the field of efficient neural networks (Bengio et al., 2013; Li et al., 2017; Yin et al., 2019).

To tackle the discontinuity challenges inherent in training efficient neural networks, algorithms have primarily focused on adapting gradient descent algorithms to work with non-differentiable operations. Empirical techniques such as the straight-through estimator (STE) (Bengio et al., 2013) have been employed to define gradients for non-differentiable operations. However, even with the STE, the perturbation introduced by quantization has been observed to disrupt existing training recipes (Fig. 1(a)). Consequently, clipping is commonly applied to limit the signals within a small range to prevent divergence. Despite these techniques, training quantized networks remains restricted to specific precisions (Rastegari et al., 2016; Courbariaux et al., 2016; Lin et al., 2023; Liu et al., 2021a) or models (Zhang et al., 2022; 2023). When moving to low precisions, these approaches usually also make changes to model architectures and recipes, such as inserting extra normalizations(Zhang et al., 2022; Wang et al., 2023), changing learning rates(Wang et al., 2023), replacing optimizers (Liu et al., 2021a), keeping several layers unquantized (Abdolrashidi et al., 2021; Liu et al., 2021a) or through fine-tuning (McKinstry et al., 2018). It is unclear if these techniques are effective and flexible enough in the large generative AI models.

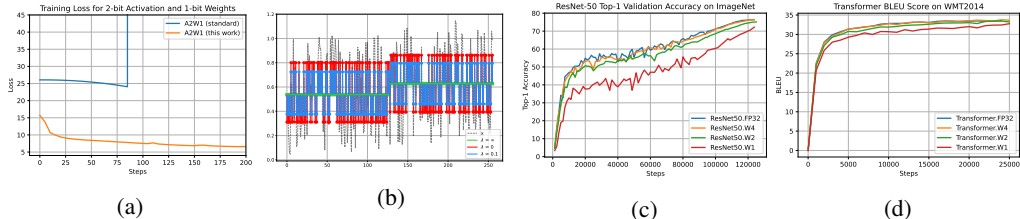

(a)          (b)          (c)          (d)

Figure 1: (a) Our approach consistently trains models at ultra-low precision levels where state-of-the-art quantization algorithms often fail due to divergence. (b) Our method decomposes the quantized signal into a noise-free, piecewise constant backbone (green) and a non-smooth component (red) capturing the perturbed signal. The non-smooth component is suppressed (blue) when combined with the backbone, ensuring training stability even at arbitrary precision and sparsity levels. We demonstrate our approach's effectiveness by training ResNet-50 (c) and Transformer (d) models at various weight precision levels (down to 1-bit), achieving state-of-the-art results. Activation is quantized to 4-bit in (c, d). Our robust training allows for exploring performance trade-offs across different precision levels. (See Section 4 for results with other precision).

In contrast to the intricate and often empirically tuned techniques of prior methods, we adopt another approach that formulates discontinuous operations as the introduction of perturbations (Golub & Van Loan, 2013). We address this challenge directly by suppressing the effects of these perturbations, effectively denoising the signal. Our approach comprises three fundamental steps:

1. Affine Transform for Quantization (Sec. 3.1): An initial affine transform $f$ scales the input signal without introducing additional operations (e.g., clipping).

2. Perturbation Injection (Sec. 3.2): A controlled perturbation $\delta$ is injected into the signal, precisely modeling the effect of quantization.

3. A Denoising Affine Transform for Reconstruction (Sec. 3.3): A key innovation of our approach is the introduction of another affine transform $g$ that effectively reconstructs the original signal while mitigating quantization noise.

Our method offers several key advantages: 1. Continuous control over quantization noise: This ensures stable model training and prevents divergence. 2. Graceful degradation: Under extreme noise, our approach seamlessly transitions to a lower-resolution (through averaging) model, guaranteeing a performance lower bound. 3. Compatibility: Our method works with existing architectures and training recipes, eliminating the need for extensive modifications or hyperparameter tuning.

These innovations enable the development of diverse, efficient neural networks for various applications. We demonstrate the effectiveness of our approach by training sparse models and temporal binary neural networks. Our key contributions include:

- A simple, robust, and universal solution for training quantized neural networks.
- A novel theoretical perspective and practical solution for training networks with discontinuous operations, based on decomposition and denoising.
- State-of-the-art results with ultra-low precision models, using standard architectures and training recipes.
- Easy adaptation to train sparse neural networks.
- A novel shortcut formula for computing quantized matrix multiplication.
- Successful training of temporal binary neural networks, demonstrating its potential for bridging the gap between artificial and biological neural networks.

## 2 MOTIVATIONS

### 2.1 A COMPREHENSIVE PICTURE OF MODEL EFFICIENCY

While extreme quantization and sparsification can significantly reduce storage and computational requirements, they also introduce the risk of quality degradation. Existing studies on quantization and sparsity primarily focus on showcasing the effectiveness of specific implementations through

meticulous tuning. While these studies provide valuable insights, they fall short of providing a comprehensive understanding of the performance trade-offs involved in applying these techniques to extreme levels. The absence of a robust, universal algorithm for training quantized and sparse neural networks has hindered an accurate comparison of different approaches and has limited the exploration of the full potential of these techniques. To address these limitations, a more comprehensive and rigorous approach is needed to evaluate the trade-off between efficiency and quality. This necessitates the development of a universal approach that can effectively manage extreme quantization and sparsity levels (Fig. 1(c,d)).

## 2.2 A Biophysical Basis for Sparse Quantization

One of the most significant achievements in 20th-century neural physiology was the development of the Hodgkin–Huxley model for modeling animal neural networks (Hodgkin & Huxley, 1952; Dayan & Abbott, 2005). This model unveiled the intriguing fact that animal neural networks primarily consist of brief electrical impulses, commonly known as spikes in their activity patterns. Despite this critical insight, the question of how animals efficiently learn and process information through these spike trains continues to pose a profound and unresolved challenge. Since the rise of regular neural networks, researchers have strived to find a more biophysically plausible approach to artificial intelligence through spiking neural networks (Yamazaki et al., 2022; Lee et al., 2016; Tavanaei et al., 2019; Gallego et al., 2020). However, the lack of differentiability in spiking neural networks prevents the straightforward application of gradient descent algorithms, the foundation of modern neural networks. Consequently, these networks have limited application and popularity. The development of universal training algorithms for quantized and sparse neural networks could open up new avenues in the field of spiking neural networks.

## 3 Methods

We primarily focus on explaining our formulation for quantization, as it is more widely supported by modern hardware and offers a broader range of applications. Subsequently, we demonstrate how our formulation can be extended to sparsification. Our method's flexibility enables it to address quantization and sparsification individually or together, allowing users to apply it to model weights and activations independently or in combination.

### 3.1 Affine Transform for Quantization

We begin our study by using min-max scaling to move and scale the floating-point vector $\boldsymbol{x}$ to the desired range, e.g. $[0, 2^{bits} - 1]$. We formulate with this range for simplicity. The range can be shifted if signed integers are more compatible with specific hardware. This affine transform is implemented using standard functions in neural network libraries for backpropagation training.

$$f(\boldsymbol{x}) = \frac{\boldsymbol{x} - x_{min}}{x_{max} - x_{min} + \epsilon} \cdot (2^{bits} - 1) \tag{1}$$

### 3.2 Perturbation Injection

Quantization reduces the number of bits used to represent $f(\boldsymbol{x})$ by rounding to the nearest integer. This rounding operation introduces discontinuities at half-integer values, rendering it non-differentiable. These discontinuities can lead to training difficulties, as neural networks rely on gradients for learning. Most existing methods for training neural networks with non-smooth operations employ the straight-through estimator (STE) (Bengio et al., 2013). The STE provides a technique for defining gradients for non-differentiable operations. However, even with the STE, training algorithms may diverge when moving to lower bits (Fig. 1(a)).

Building upon prior research (Golub & Van Loan, 2013; Bengio et al., 2013; Lee et al., 2016), we model the impact of these operations as perturbations within the network, recognizing their role in causing training instability. Specifically, quantization of $f(\boldsymbol{x})$ can be modeled as injecting a bounded perturbation:

$$\boldsymbol{q} = f(\boldsymbol{x}) + \boldsymbol{\delta}, \tag{2}$$

where $\boldsymbol{\delta} = \text{round}(f(\boldsymbol{x})) - f(\boldsymbol{x}), \boldsymbol{\delta}_i \in [-0.5, 0.5]$. We avoid empirical operations such as clipping, which can introduce larger perturbations and degrade model performance. By directly incorporating the quantization noise into the model, our method maintains signal fidelity and achieves superior results compared to traditional clipping-based approaches. Combining with the first affine transform (Eq. 1), our quantization can be found in Fig. 2.

### 3.3 DENOISING AFFINE TRANSFORM FOR RECONSTRUCTION

A core innovation of our approach is the introduction of an additional affine transformation $g$, designed to be resilient to perturbations. This enhances the robustness of quantized neural networks, and remarkably, we find that standard training methods used for full-precision models can still effectively converge even when perturbations are present.

Quantization algorithms typically introduce a one-dimensional affine or linear transform, often referred to as dequantization, to approximate the original vector $\boldsymbol{x}$. To streamline computations, most practical implementations simply invert the scaling involved in Eq. 1 (Abdolrashidi et al., 2021; Zhang et al., 2022; 2023) or minimize the $L_2$ reconstruction error (Rastegari et al., 2016; Dettmers et al., 2023). However, focusing solely on minimizing this approximation error may overlook the crucial challenge of training neural networks to be robust against perturbations (Fig. 1(a)).

Our approach, counterintuitively, increases the approximation error but naturally resolves this long-standing issue. We formulate the reconstruction as a ridge regression problem, introducing a regularization factor $\lambda$ and solving for two parameters:

$$\min_{a,b} \frac{1}{2N}||a \cdot \boldsymbol{q} + b - \boldsymbol{x}||^2 + \frac{\lambda}{2}a^2 \tag{3}$$

Here $N$ is the dimension/length of $\boldsymbol{x}$. Taking the derivative with respect to $b$, and setting to zero yields the following equation: $\frac{1}{N}\sum_i(b + a\boldsymbol{q}_i - \boldsymbol{x}_i) = 0$.

Solving for $b$ yields:

$$b = \overline{\boldsymbol{x}} - a\overline{\boldsymbol{q}} \tag{4}$$

And from setting the derivative with respect to $a$ to zero: $\frac{1}{N}\sum_i \boldsymbol{q}_i(a\boldsymbol{q}_i + b - \boldsymbol{x}_i) + \lambda a = 0$. Simplifying the equation, we obtain:

$$\overline{\boldsymbol{q}^2}a + \overline{\boldsymbol{q}}b - \overline{\boldsymbol{x}\boldsymbol{q}} + \lambda a = 0 \tag{5}$$

Substituting Eq. 4 into Eq. 5: $\overline{\boldsymbol{q}^2}a + \overline{\boldsymbol{q}}\overline{\boldsymbol{x}} - \overline{\boldsymbol{q}}^2 a - \overline{\boldsymbol{x}\boldsymbol{q}} + \lambda a = 0$.

We arrive at the solution for $a$:

$$a = \frac{Cov_{xq}}{Var_q + \lambda} \tag{6}$$

Substituting $a$ and $b$ back into the ridge regression yields the reconstructed vector, representing the quantized version of $\boldsymbol{x}$:

$$\boldsymbol{r} = g(\boldsymbol{q}) = a \cdot \boldsymbol{q} + b = \frac{Cov_{xq}}{Var_q + \lambda}(\boldsymbol{q} - \overline{\boldsymbol{q}}) + \overline{\boldsymbol{x}} \tag{7}$$

This affine transformation can also be seamlessly implemented using fundamental operations readily available in neural network libraries (Fig. 3). Quantization and reconstruction can be represented as a straightforward addition of $\boldsymbol{\delta}$ between the two affine transform operations: $r = g(f(\boldsymbol{x}) + \boldsymbol{\delta})$. During actual computations, $f(\boldsymbol{x}) + \boldsymbol{\delta}$ is cast to the appropriate data type.

#### 3.3.1 A NOVEL VIEW OF THE QUANTIZED SIGNAL

The reconstruction from Eq. 7 provides a novel decomposition of the quantized signal $\boldsymbol{r}$ into a smooth component, $\overline{\boldsymbol{x}}$, and a non-smooth component, $a(\boldsymbol{q} - \overline{\boldsymbol{q}})$, drawing inspiration from detail enhancement techniques in computer vision (He et al., 2012; Farbman et al., 2008). Importantly, the bounded perturbation from quantization, $\boldsymbol{\delta}$, is entirely contained within this non-smooth component. This observation is key, as this non-smoothness directly contributes to training instability. Consequently, we can stabilize the training by suppressing this component using the parameter $\lambda$. In essence, $\lambda$ acts as a control knob, regulating the balance between signal and noise that enters the training process, as visualized in Fig. 1(b).

Our design exhibits two important properties:

**Proposition 1.** *By adjusting $\lambda$, the quantized model can be trained to converge if the training algorithm converges on a smaller scale network.*

*Proof.* As $\lambda$ approaches infinity, the perturbation can be completely suppressed, resulting in the mean value $\overline{x}$ serving as a fail-safe vector for the reconstruction of $x$. This behavior is mathematically expressed as: $\lim_{\lambda\to\infty} a = 0, \lim_{\lambda\to\infty} b = \overline{x}$. □

The structure of this smaller scale network will be explained in Sec. 3.5. In practice, we find that a small $\lambda$ provides a good trade-off between preserving the original signal and suppressing quantization noise. This leads us to discuss the other extreme case of $\lambda = 0$.

**Proposition 2.** *The regularized model evolves continuously from the original model through the control parameter $\lambda$.*

*Proof.* Setting $\lambda = 0$ (no regularization) and $\boldsymbol{\delta} = 0$ (eliminating perturbations) restores the network to its original form. Intuitively, this behavior arises from the fact that for unperturbed input data $x$, $q = f(x)$. The scaling factor $a$, calculated as $\frac{Cov_{xf(x)}}{Var_{f(x)}} = \frac{x_{max}-x_{min}+\epsilon}{2^{bits}-1}$, effectively inverts the scaling in Eq. 1. Since $a \cdot f(x) = x - x_{min}$, the reconstructed signal $r$ is then computed as $r = a \cdot (f(x) - \overline{f(x)}) + \overline{x} = x - x_{min} - \overline{x} + x_{min} + \overline{x} = x$. In practice, a very small $\lambda$ is used, the resulting model is a small change from the original network. This property suggests the full precision networks can be easily finetuned to a low precision networks. These properties highlight the robustness and flexibility of our proposed quantization method. □

### 3.3.2 THE SENSITIVITY OF THE SCALING FACTOR

By inverting the scaling factor in Equation 1, we can map the maximum perturbation of $0.5$ to a perturbation scaled at a magnitude of $\frac{x_{max}-x_{min}}{2^{bits}}$ in the original training signals. This implies that the perturbation intensity doubles with each reduction in bit precision, leading to a substantially amplified impact on the training process. This heightened perturbation has disrupted traditional training recipes (Fig. 1(a)), necessitating the introduction of a regularization term to stabilize the training process.

The analysis of the solution for $a$ is rooted in well-established principles of perturbation theory, as outlined in Golub's work (Golub & Van Loan, 2013) (Section 2.6).

$$(Var_q + \lambda)a = Cov_{xq} \tag{8}$$

When $\boldsymbol{\delta}$ is bounded, the impact of $\boldsymbol{\delta}$ on $a$ is proportional to $\kappa = \frac{1}{Var_q+\lambda}$ (Golub & Van Loan, 2013) (Eq. 2.6.4). $\lambda$ plays a crucial role in establishing an upper bound on $\kappa$. In the absence of $\lambda$, the influence of $\boldsymbol{\delta}$ on $a$ may become unbounded. This aligns with our observations that strong perturbations associated with lower precision lead to corrupted training. The introduction of $\lambda$ effectively stabilizes the training process by mitigating the impact of quantization noise and ensuring that the solution to Eq. 8 remains well-conditioned.

### 3.4 EXTENSION TO SPARSIFICATION

Similar to quantization, sparsification introduces discontinuities through the hard thresholding operation $H$. Consequently, the sparsification process can also be modeled as introducing perturbations, where $\boldsymbol{\delta} = H(x) - x$. In our experiment section we show this leads to significant gain in sparsifying the model. Since the perturbation from sparsification rarely leads to divergence, we may set $\lambda = 0$ to pass the entire perturbed signal when only sparsification is applied.

When dealing with extreme sparsity or quantization, it is noteworthy that $100\%$ sparsity, under the traditional definition, results in an all-zero vector. To address this limitation, we revisit Equation 7, which defines the reconstructed signal. Equation 7 involves adding back significant deviations to a mean vector of $\overline{x}$, ensuring that the mean $\overline{x}$ remains the foundation of the signal even at extreme sparsity levels. Therefore, sparsification towards the mean presents itself as the most intuitive approach. In this context, significance is measured by the deviation from the mean $x_i - \overline{x}$, not from zero. Consequently, extreme sparsity results in the mean $\overline{x}$ being preserved, rather than zero.

### 3.5 QUANTIZED MATRIX MULTIPLICATION

Consider the matrix multiplication $Y = X_{n \times i} \cdot W_{i \times o}$, we apply quantization along the rows of $X$ and columns of $W$ (Eq. 7). We aim to analyze the quantized matrix multiplication $\tilde{Y} = [(a^X \cdot \mathbf{1}') \odot Q^X + b^X \cdot \mathbf{1}'] \cdot [Q^W \odot (\mathbf{1} \cdot a^W) + \mathbf{1}^W \cdot b^W]$. Computing $\tilde{Y}$ typically leads to a sum of four terms. The following shortcut formula simplifies the sum into three terms.

**Theorem 1.** *The result $\tilde{Y}$, can be expressed as the sum of three terms: a quantized matrix multiplication term and two rank-1 terms. Specifically,*

$$\tilde{Y} = (a_{n \times 1}^X \cdot a_{1 \times o}^W) \odot (Q^X \cdot Q^W) + [\overline{X}_{n \times 1} \cdot \overline{W}_{1 \times o} - (a^X \odot \overline{Q^X})_{n \times 1} \cdot (\overline{Q^W} \odot a^W)_{1 \times o}] \cdot i \quad (9)$$

The code is presented in Fig. 4. The proof can be found in the appendix (Sec. A.5). In addition, we apply sub-channel quantization to split vectors along the contraction dimension into smaller, manageable chunks. Quantizing each chunk independently leads to a higher overall approximation quality than quantizing the entire vector or matrix at once. This finer-grained approach reduces perturbation, resulting in a more stable model. The resulting "fail-safe" backbone model (obtained as $\lambda \to \infty$) can be visualized as a piecewise constant function (Fig. 1(b)), where each piece is represented by its mean value. This is akin to lowering the granularity of the original model, effectively providing a trainable backbone with reduced complexity.

This sub-channel quantization approach permits low-precision block-wise calculation of the expensive matrix multiplication, followed by summation of partial results, can be implemented through batch matrix multiplication, drastically reducing the overall computational burden. While the provided reference code (Fig. 5) utilizes "fake-quantization" for clarity, the actual implementation is based on Eq. 9 to achieve performance gains (Sec. A.5).

The memory savings achieved depend on the chosen block size $B$. For instance, when $a$ and $b$ are stored as 16-bit floats, this results in an additional memory overhead of 32 bits per block. The effective number of bits per element is reduced to $32/B$. In practice we notice that even 8-bit floats deliver similar quality results. A typical block size of 128 is employed to maintain a storage overhead of less than one bit. The block size introduces a trade-off between memory savings and model accuracy. We investigate this trade-off further in our experiments.

## 4 EXPERIMENTS

To ensure consistency, we maintained a fixed block size of 128 across all experiments unless otherwise specified. Our experiments revealed that the regularization parameter $\lambda = 0.01$ consistently yielded satisfactory results. Quantization was applied to all linear transforms within the model, including convolutional layers. All models were trained from scratch, with architectures and parameters unchanged to enable a fair comparison between full-precision and quantized models. We utilize the notation "AxWy" to represent the quantization configuration of a neural network, where "A" denotes the bitwidth for activations and "W" denotes the bitwidth for weights. For instance, "A4W4" indicates that both activations and weights are quantized to 4 bits.

Perturbations can have a two-sided effect on neural network training. While small perturbations can mimic data augmentation and enhance the training process, they can also introduce unwanted variability into the training signal, especially when dealing with ultra-low bit and extremely sparse representations. This unwanted variability can prevent the model from learning effectively and slow down the model's convergence. To further assess the performance of the low-precision models, we extended our main experiments by four times the training duration. With extended training our 1-bit models remain highly competitive without any architectural change or recipe tweaking (Tables 1, 3). This demonstrates the robustness of our method and its ability to maintain model performance over extended training periods.

Our methodology stands out for its simplicity and generality, having undergone extensive testing on a diverse range of models and datasets, from small to large-scale. To showcase its efficacy, we present results on two well-established architectures applied to two widely recognized datasets. First, we demonstrate the effectiveness of our approach on ResNet-50 trained on ImageNet, achieving satisfactory performance without any empirical tuning. Subsequently, we evaluate our method

| Precision | 100 Epochs | 400 Epochs |
|---|---|---|
| A32W32 | 76.41 | - |
| A4W4 | **76.45** | - |
| A4W2 | 75.12 | 75.59 |
| A4W1 | 72.04 | 73.97 |

Table 1: Top-1 Validation Accuracy of ResNet-50 on the ImageNet Dataset.

| Method | FP32 | A4W4 | GE | PT | Clip | LB |
|---|---|---|---|---|---|---|
| AQT (Abdolrashidi et al., 2021) | 76.65 | 76.4* | Y | Y | Y | 4 |
| VS-Quant (Dai et al., 2021) | 76.16 | 75.28 | Y | Y | Y | 3 |
| FAQ (McKinstry et al., 2018) | 76.15 | 76.25 | Y | Y | Y | 4 |
| HAQ (Wang et al., 2019) | 76.15 | 76.14 | N | Y | Y | 4 |
| Ours | 76.41 | 76.45 | N | N | N | 1 |

Table 2: Comparison of top-1 accuracy for A4W4 ResNet-50. The columns represent: FP32: Full precision baseline. A4W4: Quantizing both activations and weights to 4 bits. GE: Whether gradient estimation is involved. PT: Pretraining/finetuning/calibration required. Clip: Clipping required. LB: The lowest bitwidth reported in the corresponding paper. * estimated from Fig. 1 (Abdolrashidi et al., 2021)

under the same setting on the Transformer model, which is more relevant to generative AI applications. Our low precision training consistently surpasses the full-precision training results, demonstrating its adaptability to different model architectures and tasks.

## 4.1 ULTRA-LOW PRECISION MODELS

### 4.1.1 RESNET-50 ON IMAGENET

We utilized the Flax framework to train ResNet-50 from scratch on ImageNet, employing stochastic gradient descent with an initial learning rate of 0.1, training the model for 100 epochs with weight decay of 0.0001. As shown in Table 1, the top-1 accuracy from the A4W4 configuration (76.45) surpasses the baseline (76.41) without any hyperparameter tuning. This demonstrates the effectiveness of our method in achieving competitive performance without requiring extensive optimization. We compared our results to previously reported A4W4 quantization results for ResNet-50 trained on ImageNet. Our results compare favorably to existing work, without the need for additional operations such as parameter search, fine-tuning, calibration, clipping, gradient estimation, or reinforcement learning (Table 2).

### 4.1.2 TRANSFORMER ON WMT

To evaluate the effectiveness of our method on transformer models, we employed the Flax framework to train the transformer model on two WMT2017 datasets (EN-DE, DE-EN) and subsequently assessed its performance on the corresponding WMT2014 datasets. The training process utilized the AdamW optimizer with weight decay set to 0.1 and a cosine scheduler for 25,000 steps, employing a batch size of 1024. Recognizing the known slow convergence of transformer models, we extended the training duration to 100,000 steps (Table 3). Remarkably, our low-precision results consistently surpass the full-precision baseline.

Given the prevalence of transformers in large language models, extensive research has been dedicated to quantizing transformer models. We compare our findings to other works, and ours stands out as the only method that can surpass the full-precision baseline (Table 4). This achievement highlights the unique strength of our formulation, which not only preserves signal fidelity but also benefits from regularization effects. Several recent works have explored alternative quantization approaches using different datasets, which are not included in this table. One such method is AWQ, a weight-only quantization 4-bit quantization method (Lin et al., 2023), requires retaining $1\%$ of salient weights and all activations unquantized. Their method also involves searching for an optimal scaling factor and a calibration set. Additionally, BitNet (Wang et al., 2023), presents a 1-bit quantization method for transformers. The lowest activation precision achieved in their work is 8 bits, exceeding the highest activation bit in our method. Their method also necessitates clipping, additional normalization, and recipe changes.

|  | DE-EN | | EN-DE | |
|---|---|---|---|---|
| Steps | 25k | 100k | 25k | 100k |
| A32W32 | 33.5 | 33.9 | 29.49 | 29.8 |
| A4W4 | 33.78 | 33.64 | 29.71 | **30.17** |
| A4W2 | 33.45 | **34.04** | 28.58 | 30.03 |
| A4W1 | 32.76 | 33.66 | 27.06 | 28.32 |
| A2W2 | 32.32 | 33.51 | 27.56 | 28.61 |
| A2W1 | 31.39 | 32.51 | 26 | 27.4 |
| A1W1 | 27.4 | 28.27 | 21.42 | 23.64 |

Table 3: BLEU Score of training low precision Transformers on the WMT datasets.

| Method | FP32 | A4W4 | GE | PT | Clip | LB |
|---|---|---|---|---|---|---|
| LSQ+LUQ (Xi et al., 2023) | 27.5 | 27.17 | Y | N | Y | 4 |
| Fixed-Point (Boo & Sung, 2020) | 28.48 | 26.94 | Y | Y | Y | 4 |
| GradScale (Sun et al., 2020) | 27.5 | 25.9 | Y | N | N | 4 |
| LUQ+SMP (Chmiel et al., 2021) | 27.5 | 27.25 | N | Y | Y | 4 |
| Ours | 29.49 | 29.71 | N | N | N | 1 |

Table 4: BLEU score comparison of A4W4 transformers. Columns are defined as in Table 2

### 4.1.3 BINARY TRANSFORMERS

In biological neural networks, information is transmitted via electrical impulses called action potentials, or spikes. The complex processes governing spike transmission within the nervous system have been extensively investigated (Dayan & Abbott, 2005). While binary signals can effectively represent these spike trains, the learning rule for spikes remains an open question (Yamazaki et al., 2022; Tavanaei et al., 2019).

Inspired by the temporal nature of the transformer model, we reduced the activation precision to 1-bit for all linear layers, transforming it into a temporal binary network, akin to a quasi-spiking neural network. However, unlike traditional spiking neural networks, our model doesn't simulate spike generation or consider spiking frequency.

To evaluate our approach, we assigned 1, 2, and 4 bits to the weights (Table 5). While introducing perturbations into spiking neural networks during training using backpropagation is not entirely new, our approach differs from previous attempts in several key aspects. Earlier studies have predominantly focused on introducing perturbations only at the spikes (Lee et al., 2016). In contrast, our formulation introduces perturbations during signal quantization, irrespective of the spikes. This mirrors the inherent noisiness of the learning process and aligns with the biological reality of neural networks, where noise is an intrinsic part of neural signaling. Our results showcase that these converted binary transformers remain highly competitive with full-precision counterparts.

A recent study attempted to binarize transformers (Zhang et al., 2023). Their approach included extra normalization layers, clipping, and progressive quantization during training. We compared our method to their A1W1 configuration (Table 1, BMT-6 in their work, trained with 200k steps), achieving significant improvements (Table 5, last column).

|  | DE-EN | | EN-DE | |
|---|---|---|---|---|
| Steps | 25k | 100k | 25k | 100k |
| A1W4 | 29.74 | 30.74 | 24.07 | 26.28 (-11.81%) |
| A1W2 | 28.81 | 29.81 | 23.4 | 25 (-16.11%) |
| A1W1 | 27.4 | 28.27 | 21.42 | 23.64 (-20.67%) |
| A1W1 (Zhang et al., 2023) | - | - | - | 17.87 (-32.18%) |

Table 5: BLEU Score of Transformers with binary activations on the WMT datasets. For the last column we record the drop from full precision models.

| | Sparsity | Baseline | Ours | + A4W4 | + A4W1 |
|---|---|---|---|---|---|
| DE-EN | 25% | 33.45 | 33.64 | 33.63 | 32.55 |
| | 50% | 33.38 | **33.73** | **33.94** | 31.32 |
| | 75% | 32.08 | 33.4 | 33.37 | 30.28 |
| | 90% | 29.5 | 31.94 | 32.22 | 29.31 |
| EN-DE | 25% | 29.25 | **30.03** | **29.92** | **27.58** |
| | 50% | 28.54 | 29.07 | 29.45 | 26.32 |
| | 75% | 27.25 | 28.98 | 28.73 | 25.05 |
| | 90% | 20.6 | 27.02 | 26.7 | 23.24 |

Table 6: BLEU score of training Transformers with sparse weights for 25k steps on the WMT datasets. We present the baseline result, our sparsification result, and ours with both sparsification and quantization. Sparsification contributes to regularization, we mark improved results in bold (compared with Table 3).

| Block | 32 | 128 | 512 |
|---|---|---|---|
| A1W1 | 29.71 | 28.27 | 27.14 |

Table 7: BLEU Score comparison of adjusting the block size when training the A1W1 Transformers for 100k steps on the WMT DE-EN Dataset.

## 4.2 QUANTIZATION AND SPARSIFICATION

In this section, we assess the effectiveness of our sparsification through perturbation proposal by comparing it to a baseline approach that employs a multiplicative mask, $\boldsymbol{w}_{sparse} = \boldsymbol{w} \odot \mathbf{1}_{|\boldsymbol{w}|>threshold}$, obtaining significant improvements. Additionally, we evaluate the performance of combining the proposed sparsity technique with quantization. Our findings indicate that moderate levels of sparsification perturbations introduce beneficial regularization effects during training, leading to improved BLEU scores. These findings are summarized in Table 6. Our supplementary section A.2 further explores the integration of structured sparsity, demonstrating how its combination with binarization techniques yields sub-1 bit models that achieve competitive performance.

Quantization and sparsification are both valuable approaches for compressing neural networks, but both can also introduce performance degradation. Therefore, carefully balancing these techniques is crucial to achieve the desired trade-off between model size and accuracy. Sparsification is performed within each block (of size 128) before quantization. This choice of order is made because quantized values have limited sparsity levels, making the reverse process less well-defined and potentially less effective. Striking a balance between compression and accuracy is paramount when applying quantization and sparsification techniques. Excessive application can lead to substantial performance degradation. Our experiments demonstrate that low-level sparsification perturbations are beneficial. However, as quantization levels increase, the tolerable level of sparsity decreases (Table 6).

### 4.2.1 BLOCK SIZE AND $\lambda$

In addition to precision and sparsity, block size also presents a trade-off between accuracy and efficiency. We observe that its influence is more pronounced at extremely low precision levels, such as 1-bit. While using a smaller block size can improve performance, the effective bits per element can easily exceed the original design. We provide some comparisons here. Considering this trade-off, lower precision models may not always be more efficient (Table 7). Firstly, the accuracy achieved with smaller blocks in an A1W1 models may not surpass the accuracy achieved using A2W2. The perturbations introduced during lower precision training often remain high, hindering the achievement of high quality. Optimal selection needs to be made based on the underlying hardware support and the problem of interest. Our work facilitates a comprehensive study of this trade-off.

The parameter $\lambda$ suppresses the impact of perturbations and prevents training explosion, especially during the transition to 1-bit quantization. Standard quantization approaches often omit the use of $\lambda$ in an attempt to minimize quantization error. However, this can lead to training divergence in the early stages, as demonstrated in Fig. 1(a). We compared with the classic inverse scaling-based quantization implementation in the AQT library (Abdolrashidi et al., 2021) to observe this effect. In our experiments, we observe that a wide range of $\lambda$ values yield satisfactory results (Table 8). However, our preference set on the safer side, and use $\lambda = 0.01$ for all settings. For higher precision ($\geq$ 4-bits), smaller $\lambda$ values, such as 0.0001, can be safely used to allow more signals to pass through

| $\lambda$ | 1.0 | 0.01 | 0.0001 | 0 |
|------|------|-------|---------|-----|
| A1W1 | 5.83 | 21.42 | 20.08 | NaN |

Table 8: BLEU Score comparison of adjusting the $\lambda$ when training the A1W1 Transformers for 25k steps on the WMT EN-DE Dataset.

without causing training instability. Extremely small $\lambda$ can cause numerical instability, particularly exacerbated by the inherent numerical noise introduced by neural network operations.

## 5 RELATED WORK

Neural network quantization has become a widely adopted technique for reducing memory footprint and accelerating inference time, enabling efficient deployment on resource-constrained devices (Gholami et al., 2022). While full-precision models typically store weights in floating-point format, quantized weights are represented as integers, typically using 8 bits (Dai et al., 2021; Wortsman et al., 2023; Jacob et al., 2018), 3-4 bits (Dettmers et al., 2023; Liu et al., 2021b; Abdolrashidi et al., 2021; Dai et al., 2021), or even 1 bit (Zhang et al., 2022; Liu et al., 2021a; Wang et al., 2023; Zhang et al., 2023; Courbariaux et al., 2016; Rastegari et al., 2016). In addition to quantizing weights, model activations can also be quantized to further enhance computational efficiency (Dai et al., 2021; Jacob et al., 2018; Esser et al., 2019).

Although 8-bit quantization is commonly used as a standard practice in industry (Jacob et al., 2018), achieving lower-bit quantization remains challenging and requires specialized techniques to ensure robust training. Several common techniques include: 1. Mixed precision quantization: This approach selectively assigns different bit levels to different weights, aiming to optimize the trade-off between model size and accuracy (Wang et al., 2019; Lin et al., 2023; Han et al., 2015; Défossez et al., 2021). 2. Training recipes: These techniques compensate for the discontinuities introduced by quantization by employing strategies such as sharpness-aware minimization (Liu et al., 2021b; Foret et al., 2020), state-aware optimization (Liu et al., 2021a), knowledge distillation (Kim et al., 2019), and multi-phase training (Liu et al., 2021c). 3. Quantization-friendly network architectures: This approach involves replacing original network layers with alternatives that are more amenable to quantization (Zhang et al., 2022).

In contrast to prior work, our method explicitly models quantization discontinuities as perturbations. We decompose the perturbed signal into clean and noisy components, then apply denoising to suppress the noise. This approach leads to a closed-form solution that guarantees training convergence even at extremely low bitwidths. While previous methods have also modeled quantization noise using continuous distributions (e.g., Uniform or Gaussian) for gradient estimation (Défossez et al., 2021; Ballé et al., 2016), they do not optimize the reconstruction process itself to enhance training stability.

To further reduce model footprint, researchers have been combining sparsity/pruning and quantization in a unified formulation to further compress neural networks (Park et al., 2022; Yang et al., 2020). In this paper, we extend our noise injection and denoising reconstruction theory to sparsity and argue that instead of pruning small values to zero, moving near-mean values to mean better preserves signal fidelity.

## 6 SUMMARY

Discontinuous operations such as quantization and sparsification pose a significant challenge to backpropagation training, as they introduce non-differentiability into the learning process. This non-differentiability has long been considered the Achilles heel of backpropagation, hindering the development of efficient and accurate neural network training algorithms. We address this challenge by re-framing these discontinuities as perturbations. This allows us to introduce a novel, continuous control mechanism that seamlessly handles non-differentiability during training. Our approach is expected to facilitate the deployment of large-scale neural network models on resource-constrained devices, enabling the widespread adoption of deep learning for mobile applications. Additionally, our technique holds promise for the development of biophysically plausible neural networks, which have the potential to revolutionize artificial intelligence and machine learning.

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

```
def quantize(x, bits, axis, eps=1e-8):
    # quantize to [0, 2^bits-1]
    max_value = jnp.max(x, axis=axis, keepdims=True)
    min_value = jnp.min(x, axis=axis, keepdims=True)
    scaled_x = (
        (x - min_value) / (max_value - min_value + eps)
        * (2**bits - 1)
    ) # scale to [0, 2^bits-1]
    delta_x = jax.lax.stop_gradient(jnp.round(scaled_x) - scaled_x)
    q = scaled_x + delta_x  # perturb to integers
    return q
```

Figure 2: JAX reference code for quantization: $\boldsymbol{q} = f(\boldsymbol{x}) + \boldsymbol{\delta}$.

```
def reconstruct(q, x, axis, lmd=1e-2):
    # ridge regression
    E_q2 = jnp.mean(q**2, axis=axis, keepdims=True)
    E_q = jnp.mean(q, axis=axis, keepdims=True)
    E_qx = jnp.mean(q * x, axis=axis, keepdims=True)
    E_x = jnp.mean(x, axis=axis, keepdims=True)

    Var_q = E_q2 - E_q**2
    Cov_qx = E_qx - E_q * E_x
    a = Cov_qx / (Var_q + lmd)  # b = E_x - a * E_q

    return a * (q - E_q) + E_x  # r = a * q + b
```

Figure 3: The denoising affine transform $\boldsymbol{r} = g(\boldsymbol{q})$.

## A   APPENDIX

### A.1   REFERENCE CODE

We provide the reference code for:

- $\boldsymbol{q} = f(\boldsymbol{x}) + \boldsymbol{\delta}$ (Fig. 2)

- $\boldsymbol{r} = g(\boldsymbol{q})$ (Fig. 3)

- The Shortcut formula for quantized matrix multiplication (Fig. 4)

- Quantized matrix multiplication with subchannel quantization (Fig. 5)

### A.2   TERNARY WEIGHTS VIA STRUCTURED SPARSITY

Our sparsification method flexibly handles M:N structured sparsity, enabling a class of ultra-low precision models (even below 1-bit) by combining quantization and sparsity.

First, we introduce perturbations to enforce an M:N sparsity constraint (for simplicity, we set $N = 4$ and $M \in 1, 2, 3$). Non-zero values are then quantized to $-1, 1$ by taking their sign. This structured sparsity effectively introduces an extra bin of 0, resulting in ternary weight encoding.

```
def quantized_matmul_shortcut(x, w, l_bits, r_bits, lmd=1e-2):
    q_x, a_x, _ = quant(x, bits=l_bits, axis=1, lmd=lmd)
    q_w, a_w, _ = quant(w, bits=r_bits, axis=0, lmd=lmd)

    n = x.shape[-1]
    res = a_x * (q_x @ q_w) * a_w
    res += x.sum(1,keepdims=True) @ w.sum(0,keepdims=True) / n
    res -= (a_x * q_x.sum(1,keepdims=True)) @ (q_w.sum(0,keepdims=True) * a_w) / n
    return res
```

Figure 4: The shortcut formula for quantized matrix multiplication.

```
def fake_quant(x, bits, axis, lambda_):
    q = quantize(x, bits, axis=axis)
    return reconstruct(q, x, axis=axis, lambda_=lambda_)

def quantized_matmul(x, w, bits=4, lambda_=1e-2, block=128):
    r_x=fake_quant(x.reshape(-1, block), bits=bits, axis=1,
     lambda_=lambda_).reshape(x.shape)
    r_w=fake_quant(w.reshape(-1, block, w.shape[-1]), bits=bits, axis=1,
     lambda_=lambda_).reshape(w.shape)
    return jnp.dot(r_x, r_w)
```

Figure 5: Reference code for a quantized matrix multiplication.

| Sparsity | Precision | BLEU | Bits |
|----------|-----------|-------|------|
| Dense | Binary | 32.76 | 1 |
| 25% | Binary | 32.55 | 1 |
| 1:4 | Ternary | 32.1 | 0.5 |
| 2:4 | Ternary | 32.66 | 1 |
| 3:4 | Ternary | 33.1 | 1.5 |

Table 9: BLEU Score comparison of Transformer on WMT DE-EN using A4W1 with/without structured sparsity. Ignoring the storage overhead of the reconstruction coefficients, we report the average number of bits required to store each weight parameter in the final column.

Since model weights typically have a close to 0 mean, during weights reconstruction, we reformulating the ridge regression without the bias term $b$:

$$\min_a \frac{1}{2N}||a \cdot \boldsymbol{q} - \boldsymbol{x}||^2 + \frac{\lambda}{2}a^2 \tag{10}$$

And for each quantization block of 128, we solving for the scaling factor only:

$$a = \frac{\overline{\boldsymbol{qx}}}{\overline{\boldsymbol{q^2}} + \lambda} \tag{11}$$

We applied this method to a transformer model trained on the WMT DE-EN dataset with 4-bit activations and ternary weights. Our results show that this 2:4 ternary implementation achieves comparable performance (Table 9) with dense binary weights.

Regarding storage efficiency, encoding 1:4 structured sparsity necessitates only 2 bits for every group of 4 elements (due to one non-zero position within each group). 2:4 sparsity demands 4 bits per 4-element block, resulting in storage requirements on par with dense binary weights. Importantly, the 1:4 structured sparsity design yields a remarkably efficient 0.5-bit per element model that maintains competitive performance (as shown in Table 9). Moreover, the additional zero-bin inherent to structured sparsity contributes to both diminished storage needs and enhanced outcomes for this class of models (compared with Table 6).

### A.3 LOWER PRECISION FLOATS

In practical implementations, our scaling factor and bias demonstrate resilience to lower precision representations. Empirical evidence confirms that utilizing float8 (E5M2) precision does not adversely affect the accuracy of our results. This robustness can be interpreted as introducing minor perturbations to the scaling and bias values.

Leveraging this property, we can further reduce the subchannel block size to 32 elements while maintaining a storage overhead of less than 1 bit per element. This approach strikes a compelling balance between memory efficiency and model performance.

In addition to quantizing to integer vectors, our proposed method also supports quantization to low-precision floats. In this scenario, the quantization vector $\boldsymbol{q}$ can take on values in FP4 or FP8 formats.

### A.4 CONNECTION WITH THE STRAIGHT-THROUGH ESTIMATOR

Unlike traditional straight-through estimators (STEs) that rely on defining backward gradients (Bengio et al., 2013; Yin et al., 2019), our novel approach directly incorporates discontinuous operations into the forward pass. This is effectively a forward implementation of the STE. Our denoising reconstruction process explicitly mitigates the disruptive effects of these discontinuities.

We experimented with replacing the additive noise term (Eq. 2) with a controlled multiplicative activation function: $\delta(f(\boldsymbol{x})) = \boldsymbol{s} \cdot f(\boldsymbol{x})$ where $\boldsymbol{s} = stop\_gradient(\frac{\boldsymbol{q}}{f(\boldsymbol{x})})$ represents element-wise precomputed scaling values. This approach aimed to eliminate explicit gradient scaling during backpropagation while incorporating quantization directly into the activation function. However, our experiments did not reveal any significant performance gains resulting from this modification.

#### A.4.1 LIMITATIONS

As demonstrated in Section 4.2.1(Table 7), quantization algorithms generally achieve the highest accuracy with small sub-channel quantization blocks. However, the data reshaping required for these blocks can create a performance bottleneck, especially in low-precision settings where integer matrix multiplications are relatively fast. Furthermore, the ridge regression computations add additional overhead. To achieve an efficient implementation in practice, careful low-level optimization and consideration of the underlying hardware capabilities will be crucial.

### A.5 THE SHORTCUT FORMULA FOR QUANTIZED MATRIX MULTIPLICATION

We provide two proofs for the shortcut formula for quantized matrix multiplication (Theorem 1).

#### A.5.1 PROOF THROUGH RANK-1 APPROXIMATION

Matrix multiplication can be decomposed as:

$$
\begin{aligned}
Y =& X_{n\times i} \cdot W_{i\times o} \\
=& (X - \overline{X}_{n\times 1} \cdot \mathbf{1}^{'} + \overline{X}_{n\times 1} \cdot \mathbf{1}^{'}) \cdot (W - \mathbf{1} \cdot \overline{W}_{1\times o} + \mathbf{1} \cdot \overline{W}_{1\times o}) \\
=& (X - \overline{X} \cdot \mathbf{1}^{'}) \cdot (W - \mathbf{1} \cdot \overline{W}) + \overline{X} \cdot \mathbf{1}^{'} \cdot (W - \mathbf{1} \cdot \overline{W}) - (X - \overline{X} \cdot \mathbf{1}^{'}) \cdot \mathbf{1} \cdot \overline{W} + \overline{X} \cdot \mathbf{1}^{'} \cdot \mathbf{1} \cdot \overline{W} \\
=& (X - \overline{X} \cdot \mathbf{1}^{'}) \cdot (W - \mathbf{1} \cdot \overline{W}) + \overline{X}_{n\times 1} \cdot \overline{W}_{1\times o} \cdot i
\end{aligned}
\tag{12}
$$

If we apply scalar quantization (signed integers together with Eq. 11) to the first term:

$$
\tilde{Y} = (a_{n\times 1}^{X} \cdot a_{1\times o}^{W}) \odot (Q^{X} \cdot Q^{W}) + \overline{X}_{n\times 1} \cdot \overline{W}_{1\times o} \cdot i
\tag{13}
$$

Eq. 13 has two parts: a smooth part that is a rank-1 approximation of the matrix multiplication, the non-smooth part is adding details to the approximation. Theorem 1 states that the affine transform $g$ (Eq. 7) only adds another rank-1 correction term.

We apply the affine quantization to Eq. 12. Let the mean subtracted matrices be $X_0 = X - \overline{X}_{n\times 1} \cdot \mathbf{1}^{'}$ and $W_0 = W - \mathbf{1} \cdot \overline{W}_{1\times o}$. When equations 1, 4, 6 are applied for the quantization and reconstruction, we can be easily find that $a^{X} = a^{X_0}, b^{X} = b^{X_0} + \overline{X}$. Since $X_0, W_0$ are mean-subtracted, taking the mean gives 0.

*Proof.*

$$
\begin{aligned}
\tilde{Y} =& [A^{X} \odot (Q^{X} - \overline{Q^{X}} \cdot \mathbf{1}^{'}) + \overline{X_0} \cdot \mathbf{1}^{'}] \cdot [(Q^{W} - \mathbf{1} \cdot \overline{Q^{W}}) \odot A^{W} + \mathbf{1} \cdot \overline{W_0}] + \overline{X}_{n\times 1} \cdot \overline{W}_{1\times o} \cdot i \\
=& (a_{n\times 1}^{X} \cdot a_{1\times o}^{W}) \odot [(Q^{X} - \overline{Q^{X}} \cdot \mathbf{1}^{'}) \cdot (Q^{W} - \mathbf{1} \cdot \overline{Q^{W}})] + \overline{X}_{n\times 1} \cdot \overline{W}_{1\times o} \cdot i \\
=& (a_{n\times 1}^{X} \cdot a_{1\times o}^{W}) \odot [Q^{X} \cdot Q^{W} - \overline{Q^{X}} \cdot \mathbf{1}^{'} \cdot \mathbf{1} \cdot \overline{Q^{W}}] + \overline{X}_{n\times 1} \cdot \overline{W}_{1\times o} \cdot i \\
=& (a_{n\times 1}^{X} \cdot a_{1\times o}^{W}) \odot (Q^{X} \cdot Q^{W}) - (a^{X} \odot \overline{Q^{X}})_{n\times 1} \cdot (\overline{Q^{W}} \odot a^{W})_{1\times o} \cdot i + \overline{X}_{n\times 1} \cdot \overline{W}_{1\times o} \cdot i
\end{aligned}
\tag{14}
$$

$\square$

### A.5.2 THE DIRECT PROOF

*Proof.* Consider quantizing the matrix multiplication: $Y_{n \times o} = X_{n \times i} \cdot W_{i \times o}$. We use $\cdot$ for the regular dot product, and $\odot$ for the Hadamard (element-wise) product. Quantized matrix multiplication computes:

$$
\begin{aligned}
\tilde{Y}_{n \times o} &= [(a^X_{n \times 1} \cdot 1^X_{1 \times i}) \odot Q^X_{n \times i} + b^X_{n \times 1} \cdot 1^X_{1 \times i}] \cdot [Q^W_{i \times o} \odot (1^W_{i \times 1} \cdot a^W_{1 \times o}) + 1^W_{i \times 1} \cdot b^W_{1 \times o}] \\
&= (a^X_{n \times 1} \cdot a^W_{1 \times o}) \odot (Q^X_{n \times i} \cdot Q^W_{i \times o}) \\
&\quad + b^X_{n \times 1} \cdot [(1^X_{1 \times i} \cdot Q^W_{i \times o}) \odot a^W_{1 \times o}] \\
&\quad + [a^X_{n \times 1} \odot (Q^X_{n \times i} \cdot 1^W_{i \times 1})] \cdot b^W_{1 \times o} \\
&\quad + b^X_{n \times 1} \cdot (1^X_{1 \times i} \cdot 1^W_{i \times 1}) \cdot b^W_{1 \times o}
\end{aligned}
\tag{15}
$$

This direct implementation results in a sum of four terms. We will now simplify this expression to the sum of a quantized matrix product and two rank-1 terms.

We begin by rewriting the second term in the sum in Equation 15, by using Eq. 4:

$$
\begin{aligned}
& b^X_{n \times 1} \cdot [(1^X_{1 \times i} \cdot Q^W_{i \times o}) \odot a^W_{1 \times o}] \\
&= (X - (a^X_{n \times 1} \cdot 1^X_{1 \times i}) \odot Q^X) \cdot 1^W_{i \times 1} \cdot 1^X_{1 \times i} \cdot (Q^W_{i \times o} \odot (1^W_{i \times 1} \cdot a^W_{1 \times o})) \cdot \frac{1}{\mathbf{1}'\mathbf{1}} \\
&= (X - (A^X \odot Q^X)) \cdot \mathbf{1} \cdot \mathbf{1}' \cdot (Q^W \odot A^W)) \cdot \frac{1}{\mathbf{1}'\mathbf{1}} \\
&= [X \cdot \mathbf{1} \cdot \mathbf{1}' \cdot (Q^W \odot A^W) - A^X \odot Q^X \cdot \mathbf{1} \cdot \mathbf{1}' \cdot (Q^W \odot A^W)] \cdot \frac{1}{\mathbf{1}'\mathbf{1}}
\end{aligned}
\tag{16}
$$

Following a similar approach, we rewrite the third term in Equation 15:

$$
\begin{aligned}
& [a^X_{n \times 1} \odot (Q^X_{n \times i} \cdot 1^W_{i \times 1})] \cdot b^W_{1 \times o} \\
&= (A^X \odot Q^X) \cdot \mathbf{1} \cdot \mathbf{1}' \cdot (W - Q^W \odot A^W) \cdot \frac{1}{\mathbf{1}'\mathbf{1}} \\
&= [A^X \odot Q^X \cdot \mathbf{1} \cdot \mathbf{1}' \cdot W - A^X \odot Q^X \cdot \mathbf{1} \cdot \mathbf{1}' \cdot (Q^W \odot A^W)] \cdot \frac{1}{\mathbf{1}'\mathbf{1}}
\end{aligned}
\tag{17}
$$

The forth term can be rewritten as:

$$
\begin{aligned}
& b^X_{n \times 1} \cdot (1^X_{1 \times i} \cdot 1^W_{i \times 1}) \cdot b^W_{1 \times o} \\
&= (X - (A^X \odot Q^X)) \cdot \mathbf{1} \cdot \mathbf{1}' \cdot (W - Q^W \odot A^W) \cdot \frac{1}{\mathbf{1}'\mathbf{1}} \\
&= [X \cdot \mathbf{1} \cdot \mathbf{1}' \cdot W - A^X \odot Q^X \cdot \mathbf{1} \cdot \mathbf{1}' \cdot W - X \cdot \mathbf{1} \cdot \mathbf{1}' \cdot (Q^W \odot A^W) + A^X \odot Q^X \cdot \mathbf{1} \cdot \mathbf{1}' \cdot (Q^W \odot A^W)] \cdot \frac{1}{\mathbf{1}'\mathbf{1}}
\end{aligned}
\tag{18}
$$

Taking into account equations 16, 17, 18, we can simplify Eq. 15 into:

$$
\begin{aligned}
\tilde{Y} &= (a^X_{n \times 1} \cdot a^W_{1 \times o}) \odot (Q^X \cdot Q^W) + [(X \cdot \mathbf{1}) \cdot \mathbf{1}' \cdot W - A^X \odot Q^X \cdot \mathbf{1} \cdot \mathbf{1}' \cdot (Q^W \odot A^W)] \cdot \frac{1}{\mathbf{1}'\mathbf{1}} \\
&= (a^X_{n \times 1} \cdot a^W_{1 \times o}) \odot (Q^X \cdot Q^W) + \overline{X}_{n \times 1} \cdot \overline{W}_{1 \times o} \cdot i - (a^X \odot \overline{Q^X})_{n \times 1} \cdot (\overline{Q^W} \odot a^W)_{1 \times o} \cdot i
\end{aligned}
\tag{19}
$$

$\square$

