# OpenReview forum: "Robust Training of Neural Networks at Arbitrary Precision and Sparsity"
_ICLR.cc/2025/Conference — Submitted to ICLR 2025_

### Official Review · Reviewer_gmQa · 2024-10-22

**Soundness:** 2
**Presentation:** 2
**Contribution:** 2
**Rating:** 3
**Confidence:** 2

**Summary:**

This work proposes a denoising affine transform method for quantization and sparsification of deep learning. This is based on ridge regression between the original vector and its quantization, which reconstructs the original signal. The proposed method enables us to train deep nets with ultra-low precision and high sparsity levels. In particular, it provides training on temporal binary networks, which has been considered challenging in previous works.

**Strengths:**

- The proposed method is quite simple and implementation-friendly. It can potentially be combined with various other architectures and  algorithms.

- It empirically works even for high quantization cases such as A1W1.

**Weaknesses:**

**Lack of Comparative Experimental Validation**

The paper demonstrates the behavior of the proposed method through various experiments, but it barely shows how much better it is compared to existing methods. Starting from Table 2, the experiments only present results in terms of accuracy, with very little insight into the stability of the learning process, leaving only the final accuracy for comparison. In particular:
- Tables 2 and 4 seem to compare results, but only for the A4W4 case, where the precision is not extremely low. Moreover, the difference in accuracy with existing methods seems negligible although there is no standard deviation provided and its significance is unclear.  As long as  the metric of performance is limited to the test accuracy, I must say that the contribution is minimal.
- The comparison for ultra-low precision (A1W1) is limited to a single cell in Table 5.

Thus, while the method's functionality is evident, it is unclear in what aspects it excels.

As I am not an expert in quantization, I cannot make a definitive judgment, but it seems that there is little prior research on precision lower than A4W4. If this is true, then the main claim of the paper should not be the proposal of a method, but rather the experimental validation at precision levels lower than A4W4, which has rarely been tested before. The paper claims state-of-the-art performance in line 94, but if there are no previous studies, it would be more appropriate to describe the setup as novel.

Furthermore, without a comprehensive evaluation of factors beyond accuracy, such as the stability against hyperparameters (e.g., learning rate), changes in training time, and memory efficiency, it seems difficult to demonstrate the advantages of the proposed method.

**Questions:**

**Ambiguous and Inaccurate Mathematical Expressions**
The readability is low, and I do not believe it is at a level suitable for acceptance.

Line 161:
Although I understand the actual algorithm, I believe the mathematical formulation of the perturbation is inappropriate. Specifically, the author defines $\delta$ by $\delta = \text{round}(f(x)) - f(x)$, and substituting this into eq.(2) makes $q = \text{round}(f(x))$. So, as a mathematical operation, adding $\delta$ is meaningless for $q$. In other words, rather than adding a perturbation, it simply appears to be stating $q = \text{round}(f(x))$.

eqs.(3-8):
Clarify the dimensionality of each variable, such as $x \in \mathcal{R}^N$. It is strange that $x$ is in bold font while $b$ is in regular font, implying it is a scalar, yet the authors use $\mathbf{x} - b$, subtracting a scalar from a vector. Additionally, there is no clear definition of $\text{Cov}_{xq}$. Is this a scalar?


**Memory efficiencty**

Compared to existing methods, is the proposed method more memory efficient? Since it introduces continuous variables $a$ and $b$, it appears that training cannot be performed on low-precision hardware alone.

---

> ### Author Response · Authors · 2024-11-12
>
> Thank you for the constructive feedback.
>
> Regarding the weaknesses:
>
> 1. We acknowledge the scarcity of research on extremely low precision training (below A4W4). Our method offers a universal, principled solution applicable to all precisions, addressing this gap. While extensive fair comparisons were possible at moderate precisions, the A1W1 regime remains largely unexplored. It is noteworthy that 1-bit computation needs to handle perturbations that are 16x stronger than 4-bit computation (L241), creating a significant roadblock to ultra-low precision computing. To the best of our knowledge, no existing method can reliably handle perturbations at this scale. Relevant works at this precision involves architectural modifications. In contrast, our approach achieves superior performance (as shown in Table 5) without such tweaks, demonstrating the effectiveness of our method.
>
> 2. A key advantage of our method is its robustness and ease of use. We found that the hyperparameters optimized for full-precision models transfer seamlessly to our low-precision setting, eliminating the need for tedious hyperparameter tuning often required by other approaches.  Furthermore, the core operations of ridge regression (means and division,  as in Figure 3) minimally affects our training speed. We also discovered a novel shortcut formula to accelerate the quantized matrix multiplication computation (Figure 4). The walltime remains comparable to full-precision training. The true benefit emerges during inference, where our method enables significant memory and computational savings due to the extremely low precision of the deployed model.
>
> Regarding the questions:
>
> 1. The $round$ function introduces discontinuities that hinder gradient-based training. Directly applying $round(f(x))$ prevents proper gradient flow to $f(x)$. Our method overcomes this by injecting controlled perturbations ($f(x)+\delta$), effectively enabling gradient-based optimization even with discontinuous rounding (refer to the code L708-709).
>
> 2. We acknowledge the use of shorthand notations in our equations. The more rigorous formulation would be $a\cdot q +b \cdot 1$. To enhance clarity, we will provide this more detailed and rigorous explanation as in Appendix Section A5. Specifically, $Cov_{xq}$ is a scalar value. The code in Figure 3 further illustrates its computation and usage.

---

### Official Review · Reviewer_dJ4u · 2024-11-02

**Soundness:** 3
**Presentation:** 3
**Contribution:** 3
**Rating:** 5
**Confidence:** 3

**Summary:**

This paper proposes a denoising affine transformation that stabilizes training under these conditions. It introduces perturbation modeling and noise reduction strategies using a piecewise constant backbone model, which ensures performance stability across varying precisions and sparsities.

**Strengths:**

1. The approach achieves state-of-the-art results with models trained at very low precisions, demonstrating robustness and effectiveness.
2. Demonstrated effectiveness on both ResNet and Transformer architectures across multiple datasets, suggesting that the method is adaptable to a variety of deep learning tasks.

**Weaknesses:**

1. In Eq (1), $\epsilon$ is not defined. How do you choose it?
2. In line 162, why do you use $\delta_i$ instead of $\delta$? Is it a typo?
3. Run-time analysis is missing. This is important for low-precision networks.
4. Actual running memory consumption is missing. This is important for low-precision networks.
5. The added denoising transformation and affine operations might introduce computational overhead, especially in cases where ridge regression is used for reconstruction, possibly slowing down training on certain hardware.

**Questions:**

1. In Table 8, why there is a NaN when $\lambda=0$?
2. It is surprising that your A4W4 results are better than A32W32 results in tables 1/2/3. (Low-precision training is better than the conventional float32-precision training). Can you please make an explanation? Usually, low-precision computation reduces the accuracy during the inference.
3. How is your method's actual running memory consumption and run-time compared to other methods?

---

> ### Author Response · Authors · 2024-11-12
>
> Regarding the weaknesses:
>
> 1. To ensure numerical stability and prevent division by zero, we introduce a small constant $\epsilon$, a common practice in numerical computation.  As demonstrated in Figure 2, we empirically set its value to $1e−8$ to balance stability and precision.
>
> 2. $\delta$ represents the vector of perturbations applied to our model parameters, with $\delta_i$ denoting the perturbation applied to the $i$-th element.
>
> 3. Runtime Analysis:  While we acknowledge the dependence of real-world runtime on hardware and specific kernel implementations, our primary focus lies in demonstrating the feasibility and robustness of extremely low-precision training.  Appendix A4.1 provides a preliminary discussion of runtime considerations.
>
> 4. Our method guarantees low inference time memory usage by design. The storage overhead introduced by the affine transformations (a, b) is strictly controlled to less than 1 bit per element. This efficiency enables deployment on resource-constrained devices without sacrificing performance.
>
> 5. Our work prioritizes achieving robustness in extremely low-precision training.  While Figure 3 illustrates our approach with ridge regression. The core operations (means and division) minimally affects the training speed. We also discovered a novel shortcut formula to accelerate the quantized matrix multiplication computation (Figure 4). The wall-time remains comparable to full-precision training. The key advantage lies in the ability to serve the trained model at extremely low precision, enabling significant memory and computational savings during inference.
>
> Replies to questions:
>
> 1. When $\lambda$ is too small, the large perturbations can degrade (observable at $1e-4$) or even corrupt the training.  Furthermore, setting $\lambda$ to 0 can lead to division by zero in Equation 6, rendering the computation undefined.
>
> 2. As explained at L371/L455, the perturbations introduced by moderate quantization and sparsification act as a form of regularization. These perturbations prevent the model from overfitting to the training data, leading to improved generalization performance.
>
> 3. Our method's design inherently guarantees low inference time memory. The storage overhead for our reported models is controlled to less than 1 bit per element, as discussed in the last paragraph of Section 3.4.

---

### Official Review · Reviewer_w4VP · 2024-11-03

**Soundness:** 2
**Presentation:** 2
**Contribution:** 2
**Rating:** 3
**Confidence:** 3

**Summary:**

The paper introduce an affine transform scheme for the quantization and sparsification of neural networks.

**Strengths:**

The paper proposes a novel affine transform to perform dequantization step of quantizer by solving a regularized least squares problem with respect to the scaling factor.

**Weaknesses:**

1. The authors appear to overstate their contributions. The primary challenge in training quantized neural networks is addressing the zero-gradient issue associated with discrete loss functions. However, if I understand correctly, the main contribution here is a de-quantization scheme used to compute the scaling factor for the quantizer, which does not directly address the zero gradient issue as what straight-through estimator does.

2. In the Abstract, it is stated that "Our solution employs a piecewise constant backbone model to ensure a performance lower bound and features an inherent noise reduction mechanism to mitigate perturbation-induced corruption." Could you clarify what the "piecewise constant backbone model" and the "performance lower bound" refer to specifically? I was unable to find details on these terms in the main text.

3. It is unclear whether the authors used quantization-aware training or post-training quantization in the experiments. Could you please clarify?

4. All experiments are conducted for group-wise quantization. To better demonstrate the advantages of the proposed method, I recommend including results for channel-wise quantization as well as testing on a broader range of neural architectures, such as vision transformers and MobileNet.

5. It would be beneficial to provide more technical details on how this method is extend to sparsification.

**Questions:**

1. What is the 'Flax framework'?

---

> ### Author Response · Authors · 2024-11-12
>
> Regarding the weaknesses:
> 1. By injecting controlled perturbations (L158, reference code L708-709) to model the impact of discontinuous operations, we ensure the existence of gradients for every element. This technique can be viewed as a forward implementation of the straight-through estimator (Appendix section A.4). These perturbations pose a significant challenge to training stability, as they can grow exponentially in lower precision scenarios (L241). This growth can destabilize the training process, particularly in ultra-low precision settings, where robust training of neural networks remains an open problem. While existing approaches rely on architectural modifications or recipe changes with limited success, we believe we are the first to identify increased perturbation as the key obstacle to ultra-low precision and sparse computation. To address this challenge directly, we introduce a novel denoising affine transformation. This transformation effectively mitigates the destabilizing effects of perturbation growth, enabling stable training even in ultra-low precision regimes.
>
> 2. The piecewise constant backbone is derived from the block-wise mean of the quantized vector/parameters. As explained in Section 3.1, our method decomposes the dequantized vector into its mean and the deviation from that mean. The mean acts as a performance lower bound. For subchannel quantization, we compute these block-wise means to form the piecewise constant backbone (L281, Figure 1b). This approach ensures a robust and efficient representation.
>
> 3. Our approach falls under the category of quantization-aware training, as it explicitly incorporates the quantization process into the training procedure. This allows the model to adapt to the quantized representation and achieve better performance.
>
> 4. Table 7 in our ablation study demonstrates the impact of larger block sizes. Our primary focus is on maintaining an amortized storage overhead of less than 1 bit per element. This ensures that the reported performance accurately reflects the capability of the claimed precision.  While we report results on two popular models to facilitate fair comparisons, we have internally validated our approach on a wider range of architectures.
>
> 5. Sparsification can be viewed as a generalization of our approach, where insignificant values are perturbed to some expected values (zero, or the mean in our work). Besides the discussion in the main text, we also have an additional section (A.2) in the appendix on utilizing structured sparsity to develop sub-1 bit models.
>
>
> Reply to question:
> FLAX is a Jax based neural network library and ecosystem.

---

### Official Review · Reviewer_HQZ7 · 2024-11-03

**Soundness:** 2
**Presentation:** 3
**Contribution:** 3
**Rating:** 5
**Confidence:** 3

**Summary:**

This paper proposes a universal framework for training neural networks at arbitrary precision and sparsity by treating quantization and sparsification as perturbations and applying a denoising affine transform to stabilize training. The method is compatible with standard architectures and recipes, achieving competitive results on ResNet-50 and Transformer models at low precision. It provides theoretical grounding through ridge regression and introduces efficient matrix multiplication for quantized models.

**Strengths:**

- A unified framework that applies to both quantization and sparsification.
- Denoising affine transform for training stability is novel. By framing quantization as a perturbation and introducing a controlled denoising mechanism, the authors provide a theoretically grounded way to mitigate the instability associated with ultra-low precision training.
- A solid mathematical foundation, especially in using ridge regression to stabilize training; analysis of the parameter sensitivity; shortcut for quantized matrix multiplication and the sub-channel quantization.
- Compatibility with standard architectures and training methods, which makes the approach practical.
- Competitive performance on ResNet-50 and Transformer, better results than the baselines (including the full precision training).

**Weaknesses:**

- The proposed denoising technique is applied to both quantization and sparsification, but sparsity deserves special attention. Sparsity has unique challenges, such as gradient instability and poor convergence behavior in sparse regimes. By framing sparsification as a perturbation, the method simplifies the impact of sparsity on training. Sparse models typically exhibit unique instability issues, such as dead weights that don’t recover during training. The paper does not address how the proposed denoising affine transform could prevent or manage this issue. More dedicated experiments and ablations would be helpful to understand the unique contribution of the proposed solution to sparse training. The BLEU scores provided in Tab 6 are insufficient to validate the claim that the proposed approach robustly supports sparse training across different tasks and architectures. The method’s utility in extreme sparsity settings, such as >90%, is not tested.
- Certain layers, like the first and last layers or specific convolutional layers, may require lower sparsity levels / higher precision than intermediate layers due to their critical role in feature extraction and final prediction. The paper does not discuss how the proposed approach could be adapted to handle these layer-specific sparsity requirements, which are essential for maintaining performance.
- How does this approach work in the fine-tuning settings, where small shifts in weights might accumulate errors?
- The authors demonstrate results on ResNet-50 and Transformer models, without testing across a broader range of architectures, such as MobileNet and EfficientNet, which are widely used in low-power / mobile applications. Also consider providing performance results on architectures that have very different structural and computational properties, eg RNNs.

**Questions:**

See weaknesses above.

---

> ### Author Response · Authors · 2024-11-12
>
> We appreciate the reviewer's insightful comments.
>
> Sparse models often suffer from instability and the presence of "dead weights" – parameters that remain ineffective due to zero gradients. Our approach effectively addresses both issues. By modeling sparsification as a form of perturbation, insignificant values are perturbed towards their expected value (the mean in our formulation). These neurons are kept alive in our perturbation formulation and they continue to receive gradients, preventing them from becoming completely inactive. Our denoising affine transform, regulated by $\lambda$, further ensures that these perturbations do not destabilize the training process. We have successfully trained models with extreme sparsity, achieving reliable convergence. While accuracy may degrade for small-scale models under extreme sparsity, this trade-off is expected given the significant reduction in model size.
> Beyond the discussion in the main text, Appendix A.2 provides further details on utilizing structured sparsity to develop sub-1 bit models, demonstrating the versatility of our approach.
>
> Importantly, our method achieves strong performance even with universal quantization applied to all layers, including the first and last layers. This contrasts with existing works (including those compared to in Table 2,5), which often keep these layers at full precision. Our goal is to reveal the true potential of low-precision models without resorting to layer-specific tweaks that might inflate accuracy at the cost of generality.
>
> In our fine-tuning experiments, replacing full-precision matrix multiplications with our quantized versions across all layers proved seamless, thanks to the denoising affine transform. We encountered no additional challenges during this process.
>
> While we present results on two popular models to facilitate fair comparison. Our primary contribution lies in providing a theoretically sound formulation for robust quantized matrix multiplication, applicable to diverse model architectures and sparsity levels.

---

> > ### Comment · Reviewer_HQZ7 · 2024-11-25
> >
> > I thank the authors for the rebuttal and for clarifying their contributions regarding sparse models. Do you plan to include the internal validation results on a broader range of architectures, as mentioned in your rebuttal, in the appendix of your submission?

---

> > > ### Author Response · Authors · 2024-11-26
> > >
> > > We appreciate your interest in seeing the broader validation results. While we acknowledge the value of these findings, we feel that a deeper exploration of quantization and sparsification across diverse architectures, particularly the nuances with MobileNets, warrants a more dedicated investigation, potentially in a separate publication.
> > >
> > > We found that MobileNets, with their inherent efficiency, present unique challenges for sparsification and quantization. LLMs generally exhibit more redundancy, making them amenable to quantization, even at extremely low bit widths.
> > >
> > > In the meantime, we encourage readers to leverage our JAX code as a robust and general baseline and explore its application on their models of interest. The code is designed to be readily adaptable, requiring no significant modifications.

---

### Author Response · Authors · 2024-11-14

We thank the reviewers for their valuable feedback, which has helped us strengthen our manuscript. Our work makes a significant contribution by providing a practical and effective solution that enables robust quantization and sparsification of vision and language models, even at ultra-low precision.

Crucially, we want to highlight that ultra-low precision computation is essential for the widespread adoption of neural networks. However, it presents a significant challenge: noise and perturbations grow exponentially as precision decreases. Moving from 4-bit to 1-bit computation, for example, requires handling 16x stronger noise (L241). That is why there is little reported success on extremely low precisions. Existing works on low-precision LLMs contain ad-hoc architectural changes and recipe changes. Our work is the first to identify and directly address this hurdle to ultra-low precision training. We introduce a novel and universal method that effectively stabilizes training despite these amplified perturbations, unlocking the potential of extremely low-precision LLMs. Our method is designed for seamless integration. It requires no changes to the optimizer, hyperparameters, or loss function, as demonstrated in the provided code (Figs. 2-3), which can be directly used to train neural networks at arbitrary precision.

---

### Comment · Area_Chair_i6Zi · 2024-11-24
**Reminder - Public Discussion Phase Ending Soon**

Dear PC memebers,

Thank you for your valuable comments during the review period, which raised many interesting and insightful questions. Now the discussion period is coming to a close, please take a moment to review the authors’ responses if you haven’t done so already. Even if you decide not to update your evaluation, kindly confirm that you have reviewed the responses and that they do not change your assessment.

Timeline: As a reminder, the review timeline is as follows:

November 26: Last day for reviewers to ask questions to authors.
November 27: Last day for authors to respond to reviewers.
November 28 - December 10: Reviewer and area chair discussion phase.
December 20: Meta reviews and initial decisions are due.


Thank you for your time and effort!

Best regards,
AC

---

### Meta-Review · Area_Chair_i6Zi · 2024-12-18

**Metareview:**

This paper introduces a novel perspective by treating quantization and sparsification as noise and proposing affine transformations to "denoise" and stabilize training. The reviewers acknowledged the originality of this idea and its subsequent implementation. However, they also raised substantial concerns regarding ambiguous expressions, unclear statements, and unconvincing experimental validation.

Given these significant issues, the overall ratings are consistently negative, and I recommend rejection for ICLR2025. I hope the feedback provided will assist the authors in refining their paper and further developing this promising method to ensure its potential can be fully realized.

**Additional Comments On Reviewer Discussion:**

The comments on this paper are fairly consistent. The reviewers appreciated the basic idea but highlighted numerous ambiguous expressions and unconvincing experiments. In the rebuttal, the authors were unable to effectively address these concerns or change the reviewers’ opinions.

---

### Decision · Program_Chairs · 2025-01-22

Reject